# Kinetically driven successive sodic and potassic alteration of feldspar

Gan Duan [1✉], Rahul Ram [1], Yanlu Xing[1,2], Barbara Etschmann[1] & Joël Brugger [1✉]

The dynamic evolutions of fluid-mineral systems driving large-scale geochemical transformations in the Earth's crust remain poorly understood. We observed experimentally that successive sodic and potassic alterations of feldspar can occur via a single self-evolved, originally Na-only, hydrothermal fluid. At 600 °C, 2 kbar, sanidine ($(K,Na)AlSi_3O_8$) reacted rapidly with a NaCl fluid to form albite ($NaAlSi_3O_8$); over time, some of this albite was replaced by K-feldspar ($KAlSi_3O_8$), in contrast to predictions from equilibrium reaction modelling. Fluorine accelerated the process, resulting in near-complete back-replacement of albite within 1 day. These findings reveal that potassic alteration can be triggered by Na-rich fluids, indicating that pervasive sequential sodic and potassic alterations associated with mineralization in some of the world's largest ore deposits may not necessarily reflect externally-driven changes in fluid alkali contents. Here, we show that these reactions are promoted at the micro-scale by a self-evolving, kinetically-driven process; such positive feedbacks between equilibrium and kinetic factors may be essential in driving pervasive mineral transformations.

[1] School of Earth, Atmosphere and Environment, Monash University, Clayton, Victoria, Australia. [2] Present address: Department of Earth and Environmental Sciences, University of Minnesota Twin Cities, Minneapolis, MN, USA. ✉email: gan.duan@monash.edu; joel.brugger@monash.edu

luid–rock interactions during metasomatic and/or hydrothermal processes control the rheology, porosity structure, and element redistribution within the Earth's crust[1], and also the formation of the world's sources of metals such as Cu, Au, Ag, Mo and U. In hydrothermal ore deposits, the orebody represents only a tiny volume within fluid–rock systems that are 10's to 100's of km$^3$ in scale. Many large deposits, in particular Iron Oxide Copper Gold (IOCG)[2] and Porphyry Copper-Gold[3], are associated with extensive alteration halos[4–6] that form as a result of thermodynamic disequilibrium between rock and fluid. Since Na is the most abundant cation in most deep fluids, many of the characteristic mineral reactions are driven by alkali (Na,K) exchange. The evolution from sodic to potassic alteration is a key feature of IOCG deposits. The current consensus is that this is caused by changing chemical and/or physical conditions[2,7], e.g. cooling; decompression boiling; decreasing water-to-rock ratio; and hence can be well approximated as a (near–) equilibrium system[7,8]. However, recent progress in our understanding of the mechanisms of fluid-driven mineral reactions has highlighted the significance of kinetic factors and local equilibrium in controlling the evolution of fluid-mineral systems[9]. Interfacial fluids at fluid–mineral boundaries control reaction kinetics and mineral stability, and hence the complex feedback between fluid flow, reaction progress, and reaction-induced porosity is a key driver of crustal-scale fluid-rock interaction[10–13].

Since feldspars make up more than 50% of Earth's crust, their complex compositions and textures can shed light on the thermal and alteration history of the crust[14,15]. Pioneering experiments investigated the microstructural and chemical evolution of feldspars under hydrothermal conditions[16–19], and concluded that the reactions proceeded via a fluid-driven interface-coupled dissolution-reprecipitation (ICDR) reaction mechanism, but the products of the reaction were broadly in-line with predictions from equilibrium thermodynamics. We hypothesise that the widespread temporal and spatial association between sodic and potassic alteration may be facilitated by an interplay of kinetic and equilibrium thermodynamic factors at the fluid-mineral interface. Therefore, we examined the alteration of a mixed Na-K-feldspar (sanidine) in an originally Na-only hydrothermal fluid (NaCl or NaF) in closed system experiments at isothermal, isobaric conditions.

In this work, we observed a remarkable successive sodic and potassic alteration of sanidine rather than the expected albitisation, and discovered that the replacement kinetics increased dramatically in F-rich systems. Based on characterisation of the reaction products and thermodynamic modelling of the fluid–mineral interaction, we demonstrate that at the micro-scale, sequential sodic and potassic alterations can be driven by Na-rich fluids (i.e., not much K

in the fluid is required to stabilise K-feldspar versus albite), and is promoted by a self-evolving, kinetically-driven process.

## Results

**Reaction textures and products.** We reacted sanidine, K$_{0.62(1)}$Na$_{0.33(1)}$Ca$_{0.042(6)}$AlSi$_3$O$_8$ ($n = 13$ analyses; Supplementary Data 1) in NaCl and NaF solutions. For each experiment, around 0.11 mmol sanidine and 0.15 mmol halide (NaCl or NaF) as well as 2 mmol of isotopically distinguished deionized water (natural H$_2$$^{16}$O; H$_2$$^{18}$O from Isoflex) (Supplementary Table 1) were loaded into a gold capsule (diameter: 3 mm; length: 25 mm). The sealed capsules were heated to 600 °C at 2 kbar for run durations between 1 and 5 days followed by detailed texture and chemical characterisation of reaction products (Supplementary Methods). These P-T conditions approximate the alteration/mineralisation process of the high-temperature evolution of porphyry Cu systems (350–700 °C[20]; 400–840 °C[21]) and IOCG deposits (e.g. 500–550 °C in the Mt Isa province[7]; >600 °C in the Gawler Craton[22]).

The main reaction products are albite and/or K-feldspar for both NaCl and NaF solutions (Fig. 1; Supplementary Table 1). Small amounts of biotite formed in NaCl-only solutions, whereas fluorite and ilmenite appeared in NaF-bearing solutions (Supplementary Fig. 1). Fluorite, biotite and ilmenite occur mostly along the reaction front between albite and sanidine or filling pores.

In NaCl-solutions, the overall reactions proceeded through two stages. Stage I: an albite rim replaces the outermost part of sanidine (Fig. 2A). There can be large gaps (5–20 μm wide) or a sharp interface between the reaction rim and the pristine sanidine (Fig. 2A). Stage II: a new K-feldspar appears (Fig. 2B), replacing the albite formed in stage I from the outer grain surface. The interface between the new K-feldspar and albite is sharp without noticeable micro-scale porosity. A thin discontinuous K-feldspar rim forms via partial replacement of the albite rim (Fig. 2C) over time. The new K-feldspar is characterised by higher K and lower Na contents than the starting sanidine (Fig. 2G, H, Supplementary Fig. 2). However, only small amounts of the new K-feldspar phase form, and these amounts increase only slightly over time (0.08 vol% after 3 days, and 0.41 vol% after 5 days; Fig. 1A; Supplementary Table 1). The reaction process is greatly accelerated in the F-bearing system: large amounts of K-feldspar form within only 1 day as thick rims around the parent sanidine core (20 vol% of the new K-feldspar phase; Fig. 2D–F; Supplementary Fig. 3), with only small amounts of relic albite remaining (Figs. 1B; 2D, E, I, J). Small euhedral K-feldspar grains nucleating on the surface of the sanidine seeds or growing into the solution were only observed in F-bearing experiments (Fig. 2F). Locally, K-feldspar directly replaces sanidine (Fig. 2F insert), in which case the early albite rim is fully back-reacted.

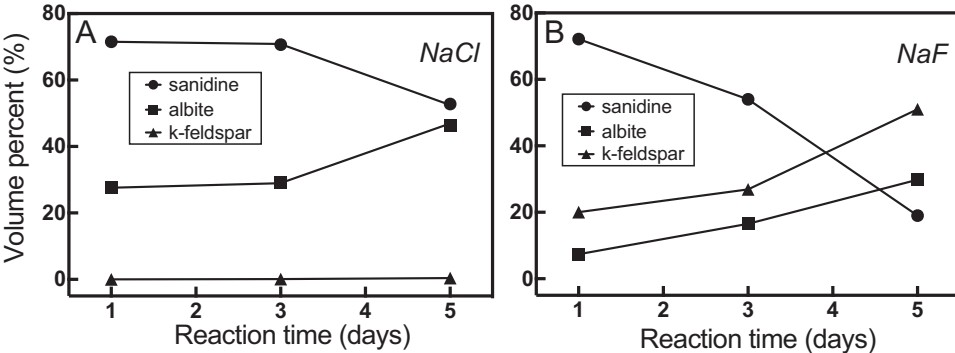

**Fig. 1 Fraction of reactant and reaction products as function of time. A** In NaCl solution. **B** In NaF solution. The analysis uncertainty is ±1%. No error bar is shown as the standard error falls within the size of the symbol. For details of the uncertainty, please check the Supplementary Information.

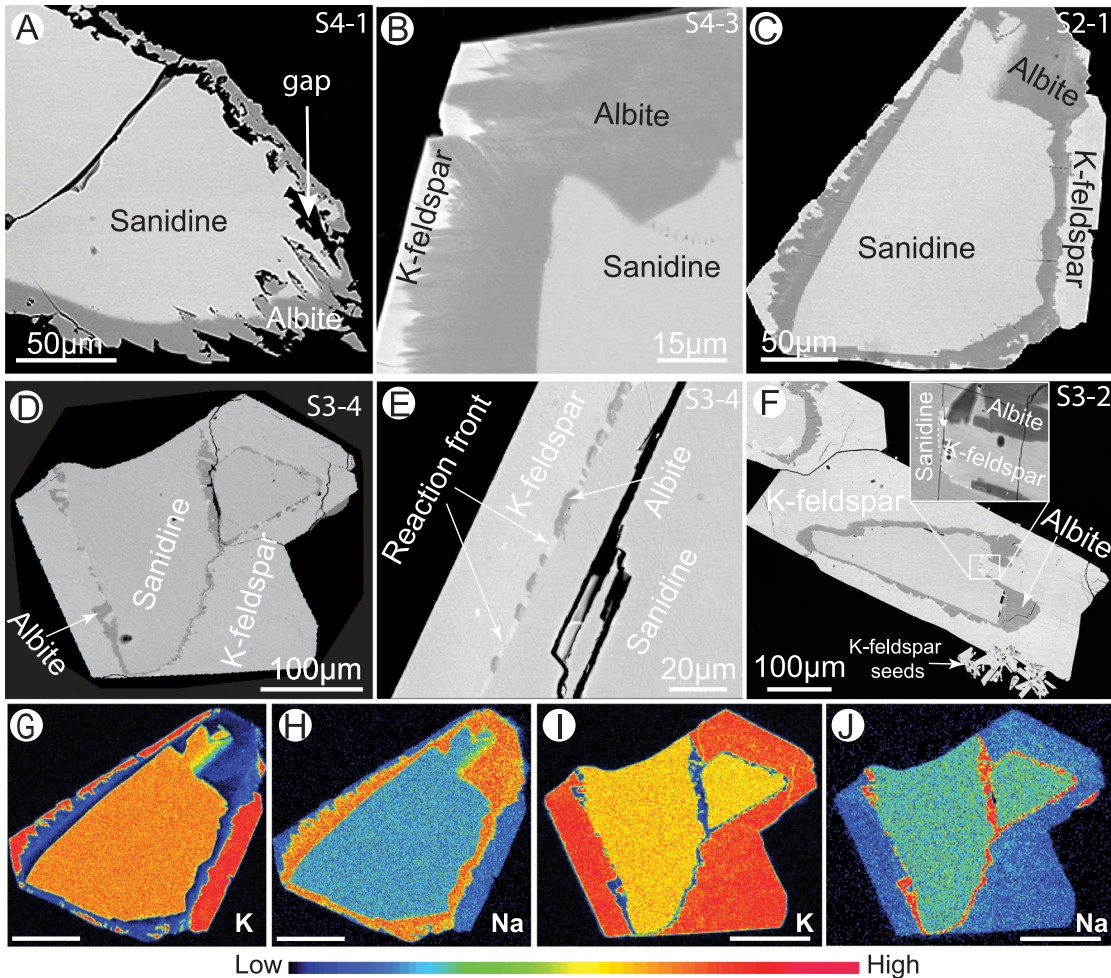

**Fig. 2 Reaction products and textures as a function of reaction time in NaCl- and NaF-solutions.** *BSE images* of reaction products from NaCl solution for 1 (**A**), 3 (**B**) and 5 days (**C**), and NaF solution for 1 (**D**, **E**) and 5 days (**F**). EMPA chemical maps of grains shown in **C** [**G**, **H**] and **D** [**I**, **J**]. Scale bars in **G**–**J** are 100 μm. The colour bar represents the contents of K or Na.

In these experiments, potassic alteration was induced without external K-input, but via dynamic evolution of fluid driven by an initial disequilibrium between mineral and Na-only fluid.

**Thermodynamic modelling**. In a chemical reaction, the composition of a product is controlled by thermodynamic and/or kinetic factors[23,24]. Here we aim to predict the system evolution under equilibrium conditions, and then use these predictions to identify potential kinetic effects that may explain the experimentally observed sequential sodic and potassic alterations. Equilibrium thermodynamic calculations are challenging since our experiments involve reactions between a complex electrolyte solution with evolving composition and mineral solid solutions characterised by a miscibility gap (Fig. 3A). Direct kinetic modelling is beyond the scope of this paper, as reliable kinetics data, such as time-resolved sampling of fluid chemistry are not available at the high temperatures and high pressures required by the reactions of interest[25].

First, we examined the fractionation of Na and K between feldspar and hydrothermal solution under static thermodynamic equilibrium conditions. All thermodynamic calculations were performed with the GEM-Selektor software package, as it has been demonstrated that this package can effectively model such complex thermodynamic conditions[26,27] (Supplementary Note). At 600 °C, the sanidine composition used in our experiments is within the miscibility gap, close to the K-feldspar equilibrium composition (Fig. 3A). The Lippmann diagram[28] in Fig. 3B depicts the aqueous

ion concentrations ($X$(Na$^+$,aq); green solutus line) in equilibrium with a given solid solution composition ($X$(Ab); red solidus line). In the presence of a Na$^+$-rich fluid ($X$(Na$^+$,aq) >0.85), the equilibrated solid is an albite-rich feldspar (Fig. 3B). At a $X$(Na$^+$,aq) of 0.85, a peritectic point is reached where the aqueous phase co-exists with two solid solutions: albite with $X$(Ab) = 0.88, and K-feldspar with $X$(Ab) = 0.35 (Fig. 3B, C). When $X$(Na$^+$,aq) decreases slightly below 0.85, the Na content of the equilibrated solid solution decreases dramatically, forming a K-feldspar with $X$(Ab) ≤0.35 (Fig. 3B). Hence, these calculations show that K-rich ($X$(Ab) of 0.13–0.22) feldspars can co-exist with Na-rich aqueous solutions (($X$(Na$^+$,aq) 0.72–0.83; Fig. 3B): K-rich fluids are not required to drive potassic alteration.

Next, progressive replacement of sanidine was modelled by aliquot titration[29] of sanidine in a fixed amount of solution at experimental conditions (2 kbar, 600 °C). Three different equilibrium scenarios were considered in terms of solution used, i.e. pure H$_2$O, NaCl-H$_2$O and NaF-H$_2$O (Fig. 4A, B). In pure H$_2$O, congruent dissolution of sanidine (Fig. 4A2) quickly results in equilibration of the solution with a feldspar of composition close to that of the titrated sanidine (Fig. 4A1). In the NaCl solution (Fig. 4B1), albite is predicted to form first, resulting in decreased Na$^+$ and increased K$^+$ in solution. When the sanidine/water molar ratio is around 0.25, two different products (Na-poor K-feldspar and a Na-rich albite) formed at the peritectic point. The NaF solution showed a similar evolution history, but the

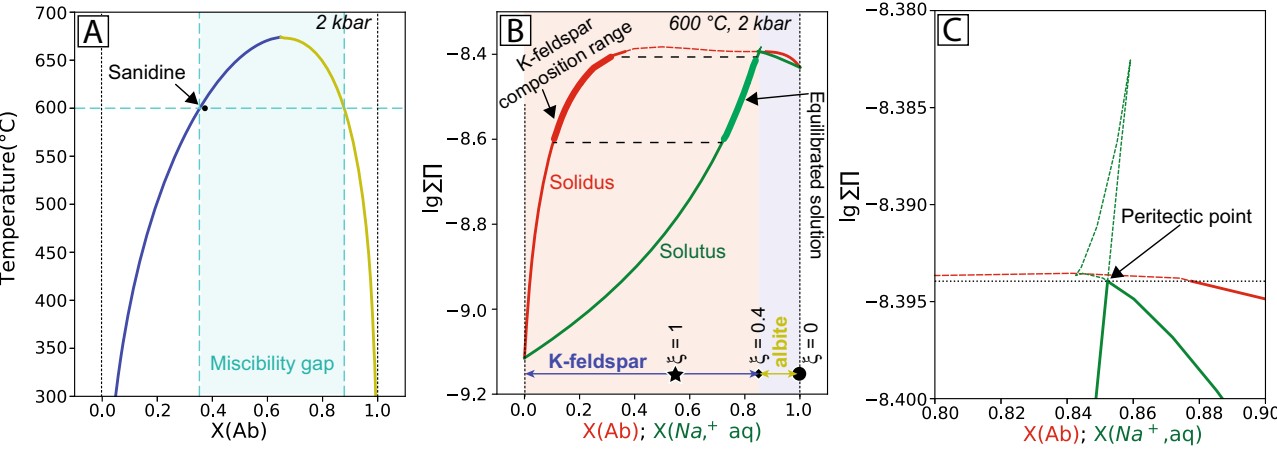

**Fig. 3 Thermodynamic modelling of the stability of Na-K-feldspars and the evolution of solids and solution composition during hydrothermal reaction.** Starting compositions of fluids and minerals are listed in Supplementary Table 2. Calculated solvus (co-existence of Na-rich and K-rich alkali feldspar) in the alkali feldspar-aqueous system (**A**) and Lippmann diagram for the aqueous–alkali feldspar–plagioclase system (**B**), as well as its upper-right fragment (**C**) calculated at $P = 2$ kbar, $T = 600$ °C. The solid and dashed lines in the Lippmann diagram are the stable and metastable segments, respectively, of the solidus and solutus curves. The peritectic point is where the aqueous phase co-exists with the two solid solution phases. $X$(Ab) is the percentage of albite end-member in each phase, and $X$(Na+,aq) the composition of the co-existing aqueous fluid ($X$(Na+,aq) = $a$(Na+)/[$a$(Na+) + $a$(K+)], where $a$ is activity). In **B** the symbol $\xi$ represents reaction progress. $\xi$ is the fraction of sanidine replaced by albite, assuming negligible dissolution of sanidine. This value highlights the solution compositions at different stages of sanidine replacement by albite, based on mass balance calculations in our experiments. This indicates that the bulk solution is expected to become saturated with respect to K-feldspar at $\xi$~0.4.

equilibrium peritectic point was reached at lower sanidine/water ratio (Fig. 4B2).

Hence, the equilibrium simulations show that unless sanidine dissolves congruently and buffers fluid composition (Fig. 4A1), it should be replaced by two feldspars at the peritectic point (Fig. 4B1-B2), with a final fluid composition with $X$(Na+,aq) = 0.85. These predictions are consistent with previous experiments[18], whereby an $Ab_{60}Or_{40}$ feldspar was replaced by coarse-grained, co-existing albite and K-feldspar upon reaction with $H_2O$/HCl solutions (Fig. 4B3).

Our experiments, however, display different final product compositions and textures (as summarised in Fig. 4C) than those predicted by equilibrium thermodynamic modelling (Fig. 4A, B). The most significant difference is that there is no evidence for peritectic co-precipitation of two feldspars in the experiments (Fig. 4C): the reaction initiates with albite nucleating on the sanidine surface in contact with the Na-rich solution, and then grows inwards via pseudomorphic replacement of sanidine. However, this newly formed albite—and some sanidine—are in turn replaced by K-rich feldspar along a separate, independent reaction front initiated from the outside of the albite rim. In theory, as the reaction products change from albite to K-feldspar, the aqueous solution in equilibrium with these feldspars should be buffered at the peritectic point. However, such co-precipitation stage was not observed in our experiments (Fig. 4C). This indicates that kinetics, rather than equilibrium, controls the reaction path and nature of products and textures in these experiments.

**Reaction mechanism.** The formation of biotite, fluorite and ilmenite co-existing with albite tallies with the complete dissolution of sanidine releasing the minor amounts of incorporated Ca, Ti and Fe. The sharp boundaries between sanidine and albite/K-feldspar and the pseudomorphic replacement are characteristic of ICDR reaction mechanism[9], which was further confirmed by experiments conducted using isotopically tagged ($^{18}O$) water. We observed sharp interfaces with large contrasts in $^{18}O$/$^{16}O$ isotopic composition between parent sanidine ($^{18}O$/$^{16}O$ < 0.003) and albite/K-feldspar (Fig. 5, Supplementary Figs. 4–6). This indicates that Si–O and Al–O chemical bonds were broken during the replacement in both Cl- and F-bearing solutions and that oxygen

exchanged widely with the solution[30], in line with the ICDR mechanism. An increase in $^{16}O$ in the framework of K-feldspar was also observed with increasing reaction time: K-feldspar formed within 1 day from a NaF solution had $^{18}O$/$^{16}O$ ratios around 2, compared to 1.4 after 5 days (Fig. 5, Supplementary Fig. 4). This suggests that K-feldspar underwent continuous dynamic re-crystallisation, which was recorded by uptake of increasing amounts of $^{16}O$ from dissolved sanidine over time. Such a dynamic recrystallisation implies that the original composition (elemental and isotopic) and texture (including porosity) of the newly formed feldspars can evolve rapidly (hours) when in contact with a fluid at 600 °C.

Fluorine facilitates the back-reaction, yet the reaction mechanisms are similar for Cl and F: sanidine dissolution, initial albitisation, followed by back-replacement by K-feldspar (Fig. 4C1). Initial sanidine dissolution results in the formation of a solution surface layer with elevated K+/Na+ ratio relative to the bulk solution; however, albite formed first, in accordance with the initial bulk fluid composition (Fig. 3B).

As the reaction rim expands, chemical exchanges between the reaction front and the bulk solution occur either through (transient) reaction-induced porosity (e.g. gap between albite and sanidine during early albite precipitation; Fig. 2A)[10], or in the absence of a connected porosity network, along the reaction interface[16,17]. In either case, the conditions at the reaction interface differ from those at the outside of the grain, but relatively fast exchange of ions between the interface and bulk fluid must happen to enable the reaction to proceed[31]. As the albite rim becomes thicker, the removal of K+ from the interface and the supply of Na+ from the bulk fluid to form albite are expected to slow down, resulting in the enrichment of K+ at the interface and rapid saturation with respect to K-feldspar (Fig. 3B). However, albite with little change in composition continues to precipitate, and K-feldspar does not nucleate at the reaction front between albite and sanidine. In other words, once albite nucleates with a particular composition, it continues to grow, irrespective of interface fluid composition. This is most likely because formation of a stable nucleus of feldspar with the new composition is statistically unlikely: atoms attaching to the growth layer edge

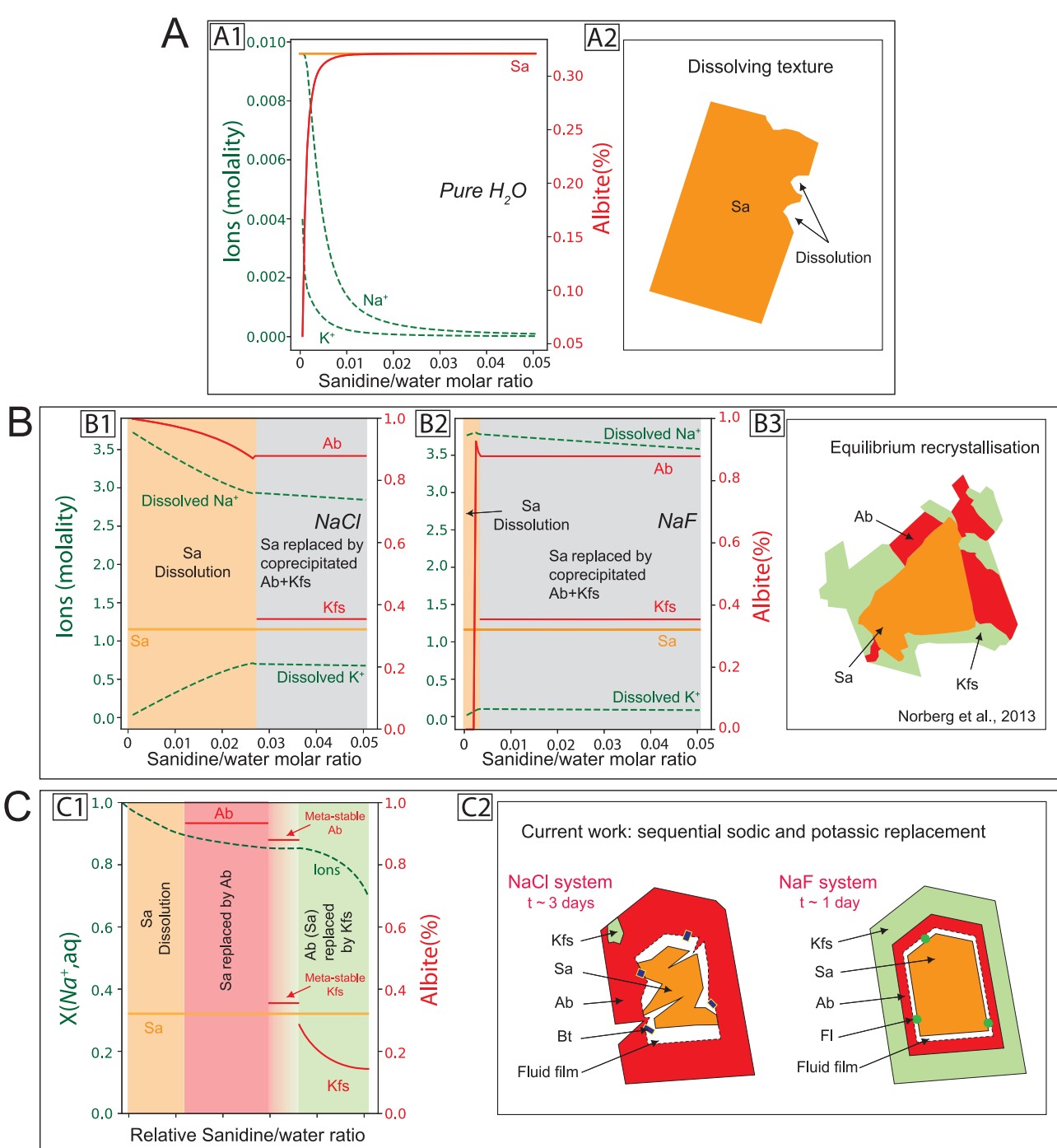

**Fig. 4 Comparison of the mineral and fluid compositions as well as the related products' texture during sanidine albitisation predicted by thermodynamic modelling with those observed experimentally.** Modelling results from (**A**) pure water; (**B**) NaCl and NaF solutions. **C** describes the extrapolated evolution of solids and fluids chemistry and the related zonation texture observed in our study. Different stages are highlighted by different background colours: orange = sanidine dissolution; grey = albite and K-feldspar co-precipitation at the peritectic point; pink = albite precipitation via replacement of sanidine; green = K-feldspar precipitation via replacement of albite (locally sanidine where albite is fully consumed). The green dashed lines represent the ions composition in solution and the red and orange solid lines represent the solid composition. (B3) shows the modelled reaction products intergrowth texture as an albite-enriched phase and a K-feldspar enriched phase co-exist with each other, which was observed in the experiment using sanidine (Ab$_{60}$Or$_{40}$) reacted with H$_2$O/HCl at 500 °C, 1000 Mpa[18]. The compositions of albite and K-feldspar (red solid lines from C1) are from electron probe analysis results (Supplementary Fig. 2). The composition of ions (green dashed line from C1) from solution is extrapolated from the Lippmann diagram in equilibrium with a given solid solution composition. The horizontal axis of (C1) shows relative increasing sanidine/water ratio from left to right. Mineral abbreviations[52]: Ab = Albite, Bt = Biotite, Fl = Fluorite, Kfs = K-feldspar, Qtz = Quartz, Sa = Sanidine.

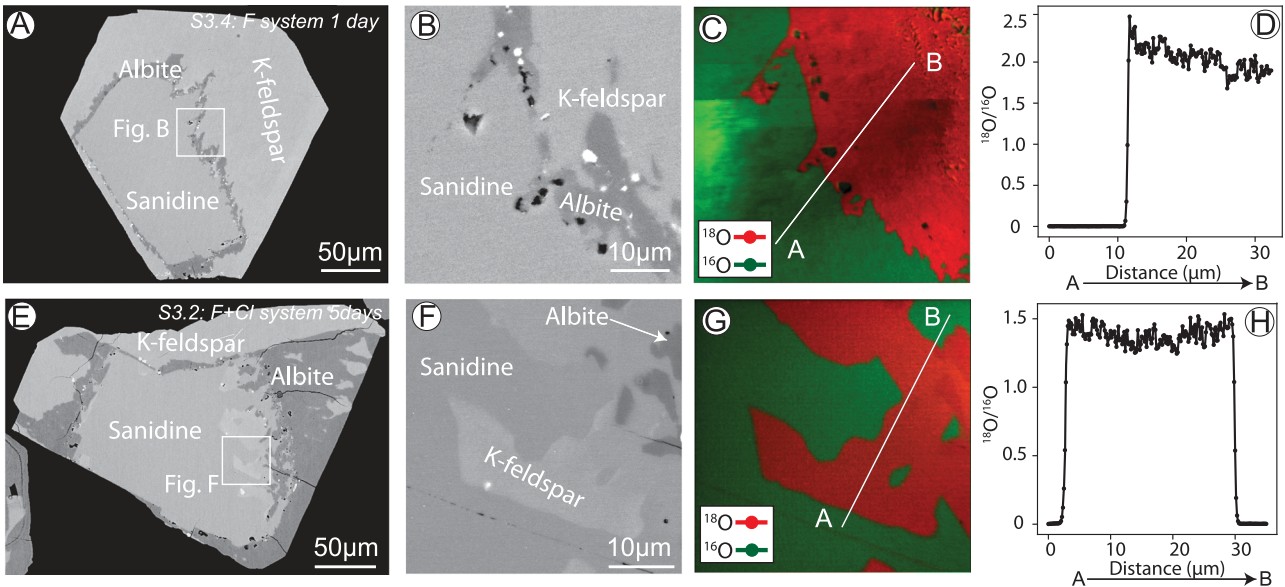

**Fig. 5 Nano-SIMS analysis of reactions products formed in [18]O-enriched solutions.** Products within 1 day (**A–D**) and 5 days (**E–H**). **A**, **B**, **E**, **F** BSE images showing the reaction products; (**C**, **G**) oxygen isotopes maps; (**D**, **H**) [18]O/[16]O line profiles of locations shown in **C**, **G**. Note the brighter field in left side of **C** is due to charging issues, which result in higher counts of ions detected (Supplementary Fig. 5).

(initial nucleation) must make two or more bonds, while only one bond is required during growth[32], and the fluid contains higher Na/K ratio than the thermodynamically stable K-feldspar.

As reaction proceeds to ~40% sanidine replacement, the bulk solution reaches the peritectic composition (Fig. 3B), and the reaction is expected to stop. Instead, albitisation proceeds (Fig. 1), until eventually, K-feldspar starts replacing albite from the outside of the grains, via a new, distinct reaction interface (Fig. 2) that is completely decoupled from the albite-sanidine interface. This behaviour is in stark contrast to the co-precipitation of albite and K-feldspar expected under equilibrium conditions. The newly formed K-feldspar is inhomogeneous and characterised by a lower Na/(Na+K) ratio (0.13–0.31 with average 0.22) than pristine sanidine (~0.35) (Supplementary Fig. 2). This wide composition range implies an evolving interfacial fluid that may become variably enriched in K[+] throughout the reaction.

Overall, in both cases (albite and K-feldspar), nucleation starts on the outside of the grain, and the daughter feldspar composition is consistent with the bulk fluid chemistry. However, each reaction then proceeds beyond the point of bulk saturation for each phase: albite replaces sanidine to an extent where the bulk solution becomes oversaturated with respect to K-feldspar, but undersaturated with respect to albite. This process then repeats itself, with nucleation of a K-rich feldspar replacing albite on the outside of the grain, leading to near-complete replacement of the earlier formed albite. We conclude that the back-reaction responsible for K-feldspar formation is the result of kinetic processes that prevent a swap in the composition of feldspar at the reaction interface: nucleation takes place at the interface between grain and bulk fluid, but dissolution and growth dominate at the reaction interface.

## Discussion

Our experiments reveal that sequential sodic and potassic alterations can be driven by a self-evolved system, whereby the overprinting of sodic alteration by potassic alteration is a kinetically triggered process without any K-input or other externally induced changes in pressure, temperature, or fluid chemistry. We further discovered that F facilitated the process, most likely by lowering the activation energy required to break Si-O and Al-O bonds to increase feldspar dissolution rates.

Moreover, we suggest that such kinetically controlled processes may be responsible for widespread grain-scale feldspar alteration in nature (e.g. Rapakivi textures), and that the positive feedback between this grain-scale mechanism and external physical and chemical drivers could be a key mechanism explaining the pervasive nature of successive sodic and potassic alteration associated with some of the World's most valuable ore deposits.

At grain scale, feldspar textures such as Rapakivi (alkali feldspar mantled by plagioclase or albite) or anti-rapakivi feldspar (plagioclase mantled by K-feldspar) are typically believed to form via a magmatic mixing process[33]. However, an external fluiddriven ICDR process was recently suggested to account for these textures[34]. Plümper et al.[10] describe one ternary feldspar grain from the Larvik batholith, Norway, that shows a similar texture as in our experiments, with pristine feldspar replaced by porous albite, which is then further replaced by K-feldspar (Fig. 2a in Plümper et al.[10]). They propose that these textures result from a fluid-driven mineral reaction. The F-rich fluid inclusions associated with some rapakivi granites (e.g. Wiborg rapakivi batholith and Kymi stock, southeast Finland[35,36]) indicate a link between high F hydrothermal fluids and Rapakivi feldspar, consistent with the increase in the kinetics of feldspar replacement observed in our experiments. Altogether, these observations show that the Rapakivi zonation textures can form in a self-evolved hydrothermal system.

Many ore-forming systems, including some porphyry Cu[4,20] and most IOCG deposits[37] are associated with large-scale (km to 10's of km) sodic alteration overprinted by potassic alteration[38]. Understanding the dynamic evolution of hydrothermal fluids in these alteration halos is the focus of numerous studies aiming to define the sources of metals, as well as identifying geochemical and mineralogical indicators for guiding mineral exploration[39,40]. In general, successive alteration styles are explained as reflecting either (i) changing compositions of fluids over time, as a result of on-going fluid-rock interactions and mineral dissolution and precipitation[38]; (ii) temperature changes in externally-derived convective systems[39,41,42]; (iii) Na-addition via interaction with evaporite-bearing wall rocks[43]; or (iv) magmatic unmixing of $H_2O-CO_2-NaCl \pm CaCl_2-KCl$ fluids caused by decreases in temperature and/or pressure[38]. One sodic-potassic alteration

zonation example is the Butte porphyry system[21]. In the K-feldspar alteration envelope, the $X(Na^+,aq)$ ranges from 0.77 to 0.85 for a hydrothermal fluid at 400-840 °C[21]. These fluid compositions are typical for magmatic fluids, that are originally rich in $NaCl$[44], as $Na^+$ and $Cl^-$ are the most common cation and anion in deep hydrothermal fluids[45]. Thermodynamic modelling shows that these fluids lie near the composition of a fluid equilibrated with albite and K-feldspar ($X(Na^+,aq)$ ~0.85 at 600 °C; Fig. 3), and can initiate proximal albitisation or potassic alteration, depending on local conditions. Albitisation results in a decreasing $X(Na^+,aq)$, a small decrease of which would result in the formation of K-feldspar. However, thermodynamic modelling suggests that potassic alteration in such a steady state system would be limited, and would feature co-existence of albite and K-feldspar, contrary to field observations of early albite and late potassic alteration. Our experiments reveal that kinetic factors acting at the reaction interface may be important in forming potassium alteration zones surrounding and/or above the sodic alteration zone in such systems. The energetic barriers that need to be overcome to change from K-feldspar nucleation to albite nucleation effectively result in the formation of K-feldspar even past the point where the fluid at the reaction front becomes supersaturated with respect to albite, as a result of increasing Na/K ratios caused by K-feldspar precipitation. This maintains a high level of disequilibrium at the reaction front, which in turns contributes to pervasive mineral replacement.

We show that K-feldspar can form from NaCl-rich fluids, i.e., small changes in fluid compositions can cause a change in fluid saturation from albite to K-feldspar (aqueous composition change around peritectic point from Fig. 3B), thereby K-rich fluid sources are not necessarily required[41], and sodic alteration followed by potassic alteration can happen even in closed systems.

Our result can explain the pervasive nature of albitisation and potassic alterations, as self-evolved processes acting at grain-scale and large-scale external drivers may work together to account for the widespread distribution of successive sodic-potassic association at both micro- and macro-scales. Field evidence suggests that large-scale successive sodic and potassic alterations are commonly associated with externally driven changes in fluid chemistry and temperature[38–43]. Plümper et al.[10] highlighted the importance of the coupling between nano-scale reaction mechanism that forms a transient reaction-induced porosity, and macro-scale fluid flow, for explaining pervasive crustal, fluid-present reactions (e.g. metamorphism; dolomitization[46]). Our results show that the internally and externally driven factors (predictable using equilibrium thermodynamics) at the macro-scale, may contribute to explaining the widespread occurrence of albitisation, as well as the common occurrence of potassic alteration overprinting this sodic alteration. It is important to note that the nano-scale factor is related to nucleation and growth at the fluid–mineral interface, and is controlled not only by external physical parameters such as pressure and temperature, but also by fluid composition, in our case the nature of halide (Cl versus F). These dynamic processes remain extremely difficult to assess on a theoretical basis, which requires further experiments as being critical in defining the mechanisms and kinetics of fluid-induced mineral reactions in the Earth's crust.

## Methods

**Experimental system**. The starting material for the experiments was a homogenous natural sanidine sample from Dellen Quarries, Niedermending, Mendig, Eifel, Rhineland-Palatinate, Germany. Some experiments were conducted in $^{18}O$-enriched water to monitor fluid-mineral exchanges[16,17].

**Sample characterisation**. The products' textures and compositions were characterised using Scanning Electron Microscopy (SEM) and Electron Micro Probe Analyser (EMPA), and the distribution of $^{18}O$ was measured semi-quantitatively by nano-Secondary Ion Mass Spectrometry (nano-SIMS) and Raman spectroscopy.

The nature of the reaction products was verified with Powder X-ray diffraction (pXRD), but Mineral Liberation Analysis (MLA) was used to quantify phase fractions, since pXRD was not able to distinguish the newly formed K-feldspar from the original sanidine. As the paper focusses on the change to the solid components during the hydrothermal alteration process, and not the solubility, fraction or speciation of feldspar minerals in fluid solutions[47–51], the bulk fluid chemistry after experiments were not analysed. Analytical methods are described in detail in Supplementary Information.

## Data availability
All data presented in this paper are included in this published article and its Supplementary information.

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

## Acknowledgements

We thank the Australian Research Council (grant DP170101893 to Joël Brugger) and the Hugh McKinstry Fund from the Society of Economic Geologist (grant SRG_20-19 to Gan Duan) for supporting this research. We acknowledge the use of the facilities and the assistance of Xiya Fang at the Monash Centre for Electron Microscopy. We also acknowledge the use of EPMA and the assistance of Nick Wilson at The Commonwealth Scientific and Industrial Research Organisation (CSIRO). We appreciate the help of Paul Guagliardo for assistance with NanoSIMS at the University of Western Australia. We are grateful to Oliver Plümper and Priyadarshi Chowdhury for comments which improved an earlier version of the manuscript.

## Author contributions

G.D. and J.B. designed the research and led the draft of this manuscript. R.R. and B.E. supported G.D. during the experimental studies and EMPA analysis. Y.X. helped with thermodynamic modelling studies. All authors contributed to the data interpretation.

## Competing interests

The authors declare no competing interests.
