## [Peer Review File · Nature Communications]

REVIEWER COMMENTS

Reviewer #1 (Remarks to the Author):

The manuscript by Duan et al. "Dynamic feldspar alteration governed by single self-evolved fluid system" that sequential sodic and potassic alterations common for porphyry and IOCG systems may be controlled by a self-evolved originally Na-enriched hydrothermal system. This is novel, and experimentally confirmed by the authors. The study contributes to the understanding the dynamic evolution of hydrothermal fluids in alteration halos. The results and conclusions of this study may be of interest to geoscientists working in hydrothermal systems, porphyries and IOCGs in particular.

The manuscript is well-written. In my opinion, the study is experimentally has no flaws, and the conclusions are justified. Very minor typo corrections can be found in two files attached.

Having said all the above, I am posing a question if Nature Communications is the right scientific journal for this study and meets our editorial requirements. The findings of the current study may be of interest to others working on porphyry and IOCG systems, but this may not be of sufficient importance to interest researchers with all various backgrounds across the geosciences.

Reviewer #2 (Remarks to the Author):

Review of manuscript NCOMMS-20-33319-T Dynamic feldspar alteration governed by single self-evolved fluid system by Gan Duan, Joël Brugger, Rahul Ram, Yanlu Xing, and Barbara Etschmann

This is a very well written, interesting paper on the role of kinetics and local solution chemistry at controlling fluid-driven mineral replacement reactions, using feldspars as model system. It provides indirect evidence of the control of nano-scale factors related to nucleation and growth from the local fluid in contact with the mineral surface on the overall evolution of the mineral assembly, challenging the current consensus on equilibrium factors (e.g. temperature and pressure) exclusively controlling most mineral replacement reactions. In this sense, the manuscript is of potential interest for a wide audience, as the observations described and conclusions obtained on the reaction mechanisms can be extrapolated to other fluid-rock interactions in a wide range of geological processes. Given my general impression of the manuscript, I think that it should be published with minor revisions. My main concern is related to the fact that, although the solids are carefully analyzed, no attempt is made to experimentally characterized the chemistry of the fluid. This is of relevance for its comparison with the thermodynamic calculations performed, and to strengthen the conclusions of the study. If this is not done, it should be at least justified why.

Other minor comments:

Lines 168-170: As the reaction rim expands, chemical exchanges between the reaction front and the bulk solution occur either through (transient) reaction-induced porosity, or in the absence of a connected porosity network as in our experiments, along the reaction interface. How can you be sure that there is no connected porosity? Even nanoscale porosity can be enough for mass transport during mineral reactions. In relation to this, molar volume changes related to the different replacement steps observed should be at less discussed.

Figure 1. Fraction of reactant and reaction products as function of time in pure NaCl (A) and NaF solutions (B). Legend should be self-explicative. How is the fraction of reactant and reaction products estimated or measured? If they are measured, which is the error in the values reported?

Reviewer #3 (Remarks to the Author):

Review of "Dynamic feldspar alteration governed by single self-evolving fluid system" by Duan et al., submitted to Nature Communications.

Mark Pearce, CSIRO Mineral Resources, Australia

The manuscript uses well constrained hydrothermal experiments to address sequential replacement of feldspars that is commonly observed in IOCG deposits globally. The experiments react K-Na feldspar with a single Na-bearing fluid at high temperature and the experimental products are examined using SEM imaging, automated mineralogy, XRD and microprobe analyses to characterise the compositions and distribution of the reaction products. Some examples are reacted with an isotopically doped water to test whether the feldspar framework is completely replaced during reaction. Results are modelled using thermodynamic equilibrium modelling of mineral and fluid compositions to explain the sequence of reaction products observed. The manuscript returns to hydrothermal mineral systems, as promised, to describe how sequential feldspar reactions might work in nature.

The experiments are interesting, well-conceived and carried out, and the clear presentation of the data complements the well written text. The inset in figure 2F is the one exception to this because it wasn't actually clear that it was an inset and the variation in contrast was a little confusing. The main conclusion of the manuscript, extrapolating from the experiments to natural systems, is that sequential replacement of albite by K-feldspar observed in IOCG systems might actually be due to a single Na-bearing fluid rather than a sequence of fluids with (often) inexplicably changing compositions or conditions. This is an attractive proposition for me personally because the sources of the multitude of fluids that are hypothesised in studies of natural systems are rarely constrained and this study provides an elegant solution to this.

While the ideas contained in the manuscript build on previous work in studying other mineral reactions and will certainly provoke a response from workers in IOCG deposits and albitisation more generally, there are several aspects that need more focus before it can be published.

First, in applying the results to IOCG systems that the manuscript could be clearer, specifically about at what scale the authors envisage these processes happening. The authors promote their model over existing models with changing fluid compositions but their model is actually one of changing fluid compositions. The difference is that the model in this manuscript has an internally changing fluid that evolved from, presumably, a fluid-buffered reservoir since there is a fluid-solid molar ratio of 20. In line 208-210, the authors suggest that the mechanism they observe might be important in forming potassic alteration zones surrounding and/or above sodic alteration zones. This suggests, especially "above", that they envisage a system larger than the grain-scale for these processes to operate on. This is similar to the ideas presented by Louise Corriveau et al where fluid continually evolve, and cool, through fluid-rock interaction so that alteration is spatially zoned (10.2113/econgeo.111.8.2045). In contrast, if the authors envisage a scenario where apparently different reactions occur at the grain-scale in a single reaction sequence then an example of this from a natural system would provide a firm link to the applied nature of the conclusions.

Second, the experiments react a NaCl fluid with Na-K-feldspar producing a peritectic fluid composition in the Na-K system that results in secondary formation of K-feldspar. While it is necessary to simplify reaction systems for experiments, the papers that the authors cite to illustrate the complexity of extant models (e.g. Oliver et al. 2004) are tackling silicate and calc-silicate rocks being replaced by albite. Without the input of K from the dissolving sanidine how would these reaction systems behave? In applying the results to mineral systems the authors suggest that K-Na fluids from granites at the peritectic would produce either sodic or potassic alteration but is this the case even if the host rocks are not K-feldspar bearing?

Third, the thermodynamic modelling illustrates the behaviour of the system outlined by Norberg et al 2013 but it does not explain the difference between their study and the current one. There is no explanation of the spatial disconnect between the albite-K-feldspar reaction and the dissolving Na-K feldspar. The model predicts that the K-feldspar should precipitate with albite when the peritectic fluid composition is reached but the K-feldspar is apparently precipitating where the fluid is likely to be most Na-rich. Is the wide reaction interface that develops significant in this? If the reaction interface is not in fact the most K-rich composition but is easily infiltrated by Na-rich fluid then the 'incubation' of K within the porosity in the albite allows K-feldspar nucleation?

Fourth, the authors suggest that in places the Na-K feldspar is reacting directly to K-feldspar but could this not be a passive microstructure where all the albite is consumed and the two K-feldspars abut one another? More importantly, if the second reaction front catches up to the first does that stop the reaction or does the

albite continue to form?

In summary, this paper presents an interesting perspective on feldspar reactions in fluid-rich systems, addressing often complex microstructures using a clear set of experiments. Strengthening the links to natural systems by clearly articulating the scales at which the authors envisage this process to be taking place, and focusing on how the second albite-K-feldspar reaction initiates at the margins would improve the manuscript and make it directly applicable to studies of large-scale hydrothermal systems.

RESPOANSE TO REVIEWERS

Reviewer #1 (Remarks to the Author):

The manuscript by Duan et al. "Dynamic feldspar alteration governed by single self-evolved fluid system" that sequential sodic and potassic alterations common for porphyry and IOCG systems may be controlled by a self-evolved originally Na-enriched hydrothermal system.

This is novel, and experimentally confirmed by the authors. The study contributes to the understanding the dynamic evolution of hydrothermal fluids in alteration halos. The results and conclusions of this study may be of interest to geoscientists working in hydrothermal systems, porphyries and IOCGs in particular.

The manuscript is well-written. In my opinion, the study is experimentally has no flaws, and the conclusions are justified. Very minor typo corrections can be found in two files attached.

We thank the Reviewer for the positive comments. We have conducted an extensive review of the entire manuscript and have now amended all the typos.

Having said all the above, I am posing a question if Nature Communications is the right scientific journal for this study and meets our editorial requirements. The findings of the current study may be of interest to others working on porphyry and IOCG systems, but this may not be of sufficient importance to interest researchers with all various backgrounds across the geosciences.

We thank the reviewer for this comment which allows us to further highlight the importance and relevance of this work to the broader geoscience community. The main conclusion of our study is that the sequential replacement of albite by K-feldspar can be due to a single Na-bearing fluid rather than a sequence of fluids with (often) inexplicably changing compositions or conditions. This is an attractive proposition as also acknowledged by Rev.3 since the sources of the multitude of fluids natural systems are

rarely constrained and our study provides an elegant solution to this. We have improved the discussion to make it clear that (i) the application of this mechanism to large-scale successive sodic and potassic alteration raises an important possibility that cannot be discounted; (ii) the newly identified mechanism provides a simple explanation for widespread grain-scale successive sodic and potassic alteration for example in Rapakivi textures; and importantly, (iii) the new kinetic effects provide a satisfactory explanation for the pervasive nature of sodic/potassic alteration, via a positive feedback between grain-scale and external drivers. We changed the title to highlight this: “*Kinetically driven successive sodic and potassic alteration of feldspar*”. We have broadened our discussion by extrapolating the results from the experiments to natural systems (either to μm grain scale and km regional scale). Thus, the findings of the current manuscript may be of interest to researchers with various backgrounds, such as economic geology, geochemistry and mineralogy.

Reviewer #2 (Remarks to the Author):

Review of manuscript NCOMMS-20-33319-T Dynamic feldspar alteration governed by single self-evolved fluid system by Gan Duan, Joël Brugger, Rahul Ram, Yanlu Xing, and Barbara Etschmann

This is a very well written, interesting paper on the role of kinetics and local solution chemistry at controlling fluid-driven mineral replacement reactions, using feldspars as model system. It provides indirect evidence of the control of nano-scale factors related to nucleation and growth from the local fluid in contact with the mineral surface on the overall evolution of the mineral assembly, challenging the current consensus on equilibrium factors (e.g. temperature and pressure) exclusively controlling most mineral replacement reactions. In this sense, the

manuscript is of potential interest for a wide audience, as the observations described and conclusions obtained on the reaction mechanisms can be extrapolated to other fluid-rock interactions in a wide range of geological processes. Given my general impression of the manuscript, I think that it should be published with minor revisions. My main concern is related to the fact that, although the solids are carefully analyzed, no attempt is made to experimentally characterize the chemistry of the fluid. This is of relevance for its comparison with the thermodynamic calculations performed, and to strengthen the conclusions of the study. If this is not done, it should be at least justified why.

We thank the reviewer for the positive assessment of this manuscript. Indeed, the reviewer correctly assesses the importance of experimentally characterising the chemistry of the fluid and its relevance for comparison to thermodynamic calculations and providing a mass balance with the solid products obtained.

Several previous papers have analysed the fluid chemistry following the completion of experiments focusing on solubility (e.g., Hemley et al., 1992; Newton and Manning, 2000), partitioning (e.g., Keppler et al., 1991) and/or speciation (e.g., Linnen et al., 1995) of various elements at high temperature and pressure conditions. They do so, using either specially designed capsules (e.g., Qi et al., 2020), or larger capsules (> 9 mm outer diameter; e.g., Hemley et al., 1992).

In the present study, we used a sealed gold capsule (outer diameter: 3.5 mm; inner diameter: 3 mm; length: 25 mm) to conduct our cold-seal experiments. The analysis of the bulk fluid composition can only be obtained of the quenched solution after conclusion of the experiment and will not be indicative of the chemistry occurring at the conditions of our experiments (600°C and 2 kbar). For these reasons, we chose to focus mainly on the solid products obtained (e.g., EMPA and Nanosims analysis) and the starting composition of the fluids, to reliably infer the various chemical changes (e.g., Na⁺, K⁺ and

oxygen isotopes) occurring during the reaction in conjunction with thermodynamic modelling of the system at experimental conditions.

We have added a clarification within the Methods section to further highlight the justification employed. To make it clearer to the readers why fluid chemistry after each experiment is not analysed, we have added the explanation in Methods parts as following:

As the paper is focused on the change in the solid component during the hydrothermal alteration process, not the solubility, fraction or speciation of feldspar minerals in fluid solutions⁴²⁻⁴⁶, the fluid chemistry after experiments is not analyzed. (L. 275-277)

42) Hemley JJ, Cygan GL, Fein JB, Robinson GR, d'Angelo WM. Hydrothermal ore-forming processes in the light of studies in rock-buffered systems; I, Iron-copper-zinc-lead sulfide solubility relations. Economic Geology 1992, 87(1): 1-22.

43) Newton RC, Manning CE. Quartz solubility in H₂O-NaCl and H₂O-CO₂ solutions at deep crust-upper mantle pressures and temperatures: 2–15 kbar and 500–900 C. Geochimica et Cosmochimica Acta 2000, 64(17): 2993-3005.

44) Keppler H, Wyllie PJ. Partitioning of Cu, Sn, Mo, W, U, and Th between melt and aqueous fluid in the systems haplogranite-H₂O–HCl and haplogranite-H₂O–HF. Contributions to Mineralogy and Petrology 1991, 109(2): 139-150.

45) Linnen RL, Pichavant M, Holtz F, Burgess S. The effect of fO₂ on the solubility, diffusion, and speciation of tin in haplogranitic melt at 850 C and 2 kbar. Geochimica et Cosmochimica Acta 1995, 59(8): 1579-1588.

46) Qi D, Behrens H, Botcharnikov R, Derrey I, Holtz F, Zhang C, et al. Reaction between Cu-bearing minerals and hydrothermal fluids at 800 C and 200 MPa: Constraints from synthetic fluid inclusions. American Mineralogist 2020, 105(8): 1126-1139.

Other minor comments:

Lines 168-170: As the reaction rim expands, chemical exchanges between the reaction front and the bulk solution occur either through (transient) reaction-induced porosity, or in the absence of a connected porosity network as in our experiments, along the reaction interface. How can you be sure that there is no connected porosity? Even nanoscale porosity can be enough for mass transport during mineral reactions. In relation to this, molar volume changes related to the different replacement steps observed should be at least discussed.

This is a valid assessment. We have deleted the phrase “as in our experiments” so as to not emphasise the role of porosity evolution which is not the primary focus of this paper. Previous research on feldspar alteration products (i.e., albite or K-feldspar) were characterized by observable micro-scale porosity (from 10 μm to 20 μm) (e.g., Niedermeier et al., 2009; Hovelmann et al., 2010; Norberg et al., 2011). This is different to our system where we cannot observe this porosity in our residue products. Norberg et al., (2010) suggested that even when porosity within the product phase during feldspar alteration exceeded 7-8%, they could not visualise an interconnected pore network (even at the sub-micron scale) in the feldspar product through TEM tomography analysis.

While three-dimensionally interconnected porosity has been assumed to enable pervasive fluid flow (e.g., Putnis 2009), replacement reactions with only minor or insignificant negative molar volume change, such as in the case of albitization of plagioclase (Hovelmann et al. 2010), or conversely, replacement reactions with significant increases in molar volume such as K-feldspathization (Niedermeier et al. 2009), indicate that relative differences in solubility between the initial and product phases alone, are not likely to generate an interconnected pore network. Therefore, without the presence of connected porosity, the reaction interface could evolve dynamically as in the present case with feldspar alteration reactions. We have amended the discussion in the Reaction

Mechanism section to further clarify these points based on the recommendation of the Reviewer as following:

For feldspar replacement reactions either with only minor or negative molar volume changes, such as albitization of plagioclase and K-feldspar^{15,18} or even significant increases in molar volume such as K-feldspathization¹⁶, there is no interconnected pore network that enables pervasive fluid flow^{15,16,18}. Thus, without connected porosity, the reaction interface, which can evolve dynamically, plays an important role in feldspar alteration reactions. (L. 171-176)

Figure 1. Fraction of reactant and reaction products as function of time in pure NaCl (A) and NaF solutions (B). Legend should be self-explicative. How is the fraction of reactant and reaction products estimated or measured? If they are measured, which is the error in the values reported?

We measured the fraction of different products obtained using Mineral liberation analysis (MLA). We made two resin blocks (>100 reacted grains) from sample S3.4 (NaCl solution for 5 days) and conducted MLA. The average number obtained from these grains were used to represent the fraction of reaction product. Based on these repeated measurements of samples obtained from the same experiment, the standard error was calculated (<1%) to represent the error of the results.

We have amended the caption of Figure 1 and detailed methods description from Supplementary data to properly highlight the error in our measurements. To make it clearer, we have added the explanation in the Caption of Figure1 and Supplementary parts as following:

The analysis uncertainty is $\pm 1\%$. No error bar is shown as the S.E. falls within the size of the symbol. For details of the uncertainty, please check the Supplementary Data. (L. 432-434).

We made two sample blocks (>100 grains) from sample S3.4 (NaCl solution for 5 days) and repeated measurement with MLA. Based on the repeated measurements, the calculated 1% standard error is used to represent the error of the results. (L. 78-81 in Supplementary Data)

Reviewer #3 (Remarks to the Author):

Review of “Dynamic feldspar alteration governed by single self-evolving fluid system” by Duan et al., submitted to Nature Communications.

Mark Pearce, CSIRO Mineral Resources, Australia

The manuscript uses well constrained hydrothermal experiments to address sequential replacement of feldspars that is commonly observed in IOCG deposits globally. The experiments react K-Na feldspar with a single Na-bearing fluid at high temperature and the experimental products are examined using SEM imaging, automated mineralogy, XRD and microprobe analyses to characterise the compositions and distribution of the reaction products. Some examples are reacted with an isotopically doped water to test whether the feldspar framework is completely replaced during reaction. Results are modelled using thermodynamic equilibrium modelling of mineral and fluid compositions to explain the sequence of reaction products observed. The manuscript returns to hydrothermal mineral systems, as promised, to describe how sequential feldspar reactions might work in nature.

We thank the Reviewer for their positive comments.

The experiments are interesting, well-conceived and carried out, and the clear presentation of the data complements the well written text. The inset in figure 2F is the one exception to this because it wasn't actually clear that it was an inset and the variation in contrast was a little confusing.

Thanks. We have modified the contrast and brightness of the insert in Fig .2F to further highlight the sanidine and K-feldspar boundary which is now clearly observed. We have also improved the labelling.

The main conclusion of the manuscript, extrapolating from the experiments to natural systems, is that sequential replacement of albite by K-feldspar observed in IOCG systems might actually be due to a single Na-bearing fluid rather than a sequence of fluids with (often) inexplicably changing compositions or conditions. This is an attractive proposition for me personally because the sources of the multitude of fluids that are hypothesised in studies of natural systems are rarely constrained and this study provides an elegant solution to this. While the ideas contained in the manuscript build on previous work in studying other mineral reactions and will certainly provoke a response from workers in IOCG deposits and albitisation more generally, there are several aspects that need more focus before it can be published.

First, in applying the results to IOCG systems that the manuscript could be clearer, specifically about at what scale the authors envisage these processes happening. The authors promote their model over existing models with changing fluid compositions but their model is actually one of changing fluid compositions. The difference is that the model in this manuscript has an internally changing fluid that evolved from, presumably, a fluid-buffered reservoir since there is a fluid-solid molar ratio of 20. In line 208-210, the authors suggest that the mechanism they observe might be important in forming potassic alteration zones surrounding and/or above

sodic alteration zones. This suggest, especially “above”, that they envisage a system larger than the grain-scale for these processes to operate on. This is similar to the ideas presented by Louise Corriveau et al where fluid continually evolve, and cool, through fluid-rock interaction so that alteration is spatially zoned (10.2113/econgeo.111.8.2045). In contrast, if the authors envisage a scenario where apparently different reactions occur at the grain-scale in a single reaction sequence then an example of this from a natural system would provide a firm link to the applied nature of the conclusions.

Indeed, Corriveau et al. suggested the formation of spatially zoned alteration through an evolved fluid-rock interaction which is consistent with our hypotheses (i/ii) postulated in the manuscript (evolution of fluid chemistry via Temperature decrease and fluid-rock interaction along a fluid flow path).

We have reworked our discussion to make clear that (i) there is strong evidence that the closed system self-evolved reaction mechanism identified is active at the grain scale in nature (example of Rapakivi textures); and (ii) the positive feedback between this grain-scale mechanism and large scale external drivers provides a novel explanation for the pervasive nature of successive sodic and potassic alteration. To make it clearer, we have added paragraphs in the Discussion part as following:

Our experiments reveal that sequential sodic and potassic alterations can be driven by a self-evolved system, whereby the overprinting of sodic alteration by potassic alteration is a kinetically triggered process without any K-input or other externally induced changes in pressure, temperature, or fluid chemistry. We further discovered that F increased the process efficiency, most likely by increasing the feldspar dissolution rates by lowering the activation energy required to break Si-O and Al-O bonds. In the following, we argue that such kinetically controlled process could be responsible for widespread grain-scale feldspar alteration in nature (e.g., Rapakivi textures), and that the positive feedback between this

grain-scale mechanism and external physical and chemical drivers could be a key mechanism explaining the pervasive nature of successive sodic and potassic alteration associated with some of the World's most valuable ore deposits. (L.194-204)

At grain scale, feldspar textures such as Rapakivi (alkali feldspar mantled by plagioclase or albite) or anti-rapakivi feldspar (plagioclase mantled by K-feldspar) are typically believed to result from a magmatic mixing process³⁰. However, an external fluid-driven ICDR process was recently suggested to be responsible for these textures³¹. Plümper et al.¹⁰ describes one ternary feldspar grain from the Larvik batholith, Norway, that shows similar texture as in our experiments, with pristine feldspar replaced by porous albite, which is then further replaced by K-feldspar (Fig. 2a in Plümper et al.¹⁰). They propose that these textures result from a fluid-driven mineral reaction. The F-rich fluid inclusions associated with some rapakivi granites (e.g., Wiborg rapakivi batholith and Kymi stock, southeast Finland^{32,33}) indicate a link between high F hydrothermal fluids and Rapakivi feldspar, consistent with the increase in the kinetics of feldspar replacement observed in our experiments. Altogether, these observations suggest that the Rapakivi zonation textures can be controlled by a self-evolved hydrothermal system. (L.205-216)

As indicated by this study, K-feldspar can form from NaCl-rich fluids, i.e. small changes in fluid compositions can cause a change in fluid saturation from albite to K-feldspar (aqueous composition change around peritectic point from Fig. 3B), thereby K-rich fluid sources are not necessarily required. Field evidence suggests that large scale successive sodic and potassic alterations are commonly associated with externally driven changes in fluid chemistry and temperature³⁵⁻⁴⁰. The self-evolved mechanism active at the grain scale in our closed system experiments may also contribute to such large-scale systems, as the positive feedback between grain-scale processes and these large-scale external drivers may work

together to explain the pervasive and widespread distribution of successive sodic-potassic association from both micro and macro scales. (L. 245-254)

30) Hibbard M. The magma mixing origin of mantled feldspars. Contributions to Mineralogy and Petrology 1981, 76(2): 158-170.

31) Mondal S, Upadhyay D, Banerjee A. The Origin of Rapakivi Feldspar by a Fluid-induced Coupled Dissolution–Reprecipitation Process. Journal of Petrology 2017, 58(7): 1393-1418.

32) Berni GV, Wagner T, Fusswinkel T, Wenzel T. Magmatic-hydrothermal evolution of the Kymi topaz granite stock, SE Finland: mineral chemistry evidence for episodic fluid exsolution. Lithos 2017, 292: 401-423.

33) Broman C, Sundblad K, Valkama M, Villar A. Deposition conditions for the indium-bearing polymetallic quartz veins at Sarvixviken, south-eastern Finland. BROMAN ET AL. INDIUM-BEARING POLYMETALLIC QUARTZ VEINS, SARVLAXVIKEN, SE FINLAND. Mineralogical Magazine 2018, 82(S1): S43-S59.

38) Corriveau L, Montreuil JF, Potter EG. Alteration facies linkages among iron oxide copper-gold, iron oxide-apatite, and affiliated deposits in the Great Bear magmatic zone, Northwest Territories, Canada. Economic Geology 2016, 111(8): 2045-2072

Second, the experiments react a NaCl fluid with Na-K-feldspar producing a peritectic fluid composition in the Na-K system that results in secondary formation of K-feldspar. While it is necessary to simplify reaction systems for experiments, the papers that the authors cite to illustrate the complexity of extant models (e.g. Oliver et al. 2004) are tackling silicate and calc-silicate rocks being replaced by albite. Without the input of K from the dissolving sanidine how would these reaction systems behave? In applying the results to mineral systems the authors

suggest that K-Na fluids from granites at the peritectic would produce either sodic or potassic alteration but is this the case even if the host rocks are not K-feldspar bearing?

Thanks. Indeed, the premise for our suggestion is that host rocks are K-feldspar bearing. We note that calc-silicate rocks and their pelitic protoliths are actually rich in K (up to 6 wt% K₂O in the samples analysed by Oliver et al. 2004, for example), sometimes with high K/Na ratios.

Third, the thermodynamic modelling illustrates the behaviour of the system outlined by Norberg et al 2013 but it does not explain the difference between their study and the current one. There is no explanation of the spatial disconnect between the albite-K-feldspar reaction and the dissolving Na-K feldspar. The model predicts that the K-feldspar should precipitate with albite when the peritectic fluid composition is reached but the K-feldspar is apparently precipitating where the fluid is likely to be most Na-rich. Is the wide reaction interface that develops significant in this? If the reaction interface is not in fact the most K-rich composition but is easily infiltrated by Na-rich fluid then the ‘incubation’ of K within the porosity in the albite allows K-feldspar nucleation?

Indeed, the aim of the equilibrium thermodynamic calculations is to highlight the fact that our experimental products and textures are not the result of an equilibrium reaction. This is important because in such a complex system with a large miscibility gap, predicting the equilibrium path is not trivial. We have improved the text and especially redrawn Fig.4 to clarify the thermodynamic modelling work and its implications. We use colours to illustrate the sequences of reactions in the model, and added a representation of the reaction actually observed in our experiments for contrast. To make it clearer, we have added a paragraph in the Thermodynamic modelling part as following:

In a chemical reaction, the composition of a reaction product can be controlled either by thermodynamic or kinetic factors^{22,23}. Here we aim to predict the system evolution under equilibrium conditions, and then use these predictions to identify potential kinetic effects that may explain the experimentally observed sequential sodic and potassic alterations. Equilibrium thermodynamic calculations are challenging since our experiments involve reactions between a complex electrolyte solution with evolving composition and mineral solid-solutions characterized by a miscibility gap (Fig. 3A). Direct kinetic modelling is beyond the scope of this paper as reliable kinetics data (e.g., time-resolved sampling of fluid chemistry) is not possible at the high temperatures and high pressures required by the reactions of interest²⁴. (L. 95-103)

22) Garvín A, Ibarz R, Ibarz A. Kinetic and thermodynamic compensation. A current and practical review for foods. Food Research International 2017, 96: 132-153.

23) Marsden SR, Mestrom L, McMillan DGG, Hanefeld U. Thermodynamically and Kinetically Controlled Reactions in Biocatalysis—from Concepts to Perspectives. ChemCatChem 2020, 12(2): 426-437.

24) Wu S-J, Cai M-J, Yang C-J, Li K-W. A new flexible titanium foil cell for hydrothermal experiments and fluid sampling. Review of Scientific Instruments 2016, 87(9): 095110

Fourth, the authors suggest that in places the Na-K feldspar is reacting directly to K-feldspar but could this not be a passive microstructure where all the albite is consumed and the two K-feldspars abut one another? More importantly, if the second reaction front catches up to the first does that stop the reaction or does the albite continue to form?

We thank the Reviewer for this is key observation. We have changed the contrast and brightness of the Fig 2F insert to make the direct contact between Na-K feldspar and K-feldspar more obvious. In the model, we provide a sanidine-albite-K-feldspar zonation texture to simplify the reaction process. However, the product's texture can be more complicated than that. For example, the product's texture showed in Fig. 5E where the

K-feldspar can not only form the rim around albite but can also exist between sanidine and albite, which may be controlled by different fluid paths. It suggests that the K-feldspar can directly react with sanidine and the reaction is largely controlled by the fluid chemistry at the reaction interface. We have improved Fig.4C to better illustrate the observed reaction sequence. To make it clearer, we have added a paragraph in the Thermodynamic modelling part as following:

the reaction initiates with stable albite nucleating on the sanidine surface in contact with the Na-rich solution, and then growing inwards. However, this newly formed albite –and some sanidine– are in turn replaced by K-rich feldspar along a separate, independent reaction front initiated from the outside of the grains. In theory, as the reaction products change from albite to K-feldspar, the aqueous solution that is in equilibrium with the solid solutions should be buffered at the peritectic point, where co-precipitation happens (Fig. 4C1). However, we did not observe the co-precipitation stage in our experiments (Fig. 4C2). This indicates that kinetics, rather than equilibrium, controls the reaction path and nature of products and textures in these experiments. We explore the details of the reaction mechanism in the following section. (L.136-145)

In summary, this paper presents an interesting perspective on feldspar reactions in fluid-rich systems, addressing often complex microstructures using a clear set of experiments. Strengthening the links to natural systems by clearly articulating the scales at which the authors envisage this process to be taking place, and focusing on how the second albite-K-feldspar reaction initiates at the margins would improve the manuscript and make it directly applicable to studies of large-scale hydrothermal systems.

Thanks. Rev.3 correctly identifies the ‘weakness’ of the paper in the scaling up to natural systems. The textures and compositions of feldspar formed via self-evolution in closed system (our experiment) could be obtained in externally driven systems as suggested by

Rev. 3. Our experiments demonstrate that sodic and potassic alterations can be controlled by a self-evolved hydrothermal process, which can then be significantly enhanced by the presence of fluorine. We have improved the discussion to make clear that (i) whether this mechanism applies to large-scale successive sodic and potassic alteration is a possibility that cannot be discounted; however, further work is required to assess the effect of mineralogical complexities, fluid flow, and time scales on the reaction. (ii) the newly identified mechanism provides a simple explanation for widespread grain scale successive sodic and potassic alteration for example in Rapakivi textures. Importantly, (iii) the new kinetic effects provide a satisfactory explanation for the pervasive nature of sodic/potassic alteration, via a positive feedback between grain-scale and external drivers. We hope that we have succeeding in capturing the spirits of Reviewer 3's insightful suggestions in this revision.

REVIEWER COMMENTS

Reviewer #2 (Remarks to the Author):

The authors have carefully addressed all the points raised in the first submission of the manuscript. I think that the manuscript should be published in its current form.

Reviewer #3 (Remarks to the Author):

Review of "Kinetically driven successive sodic and potassic alteration of feldspar" by Duan et al submitted to Nature Communications.

Mark Pearce, CSIRO Mineral Resources, Australia

I am excited to read the revised version of this manuscript and thank the authors for taking the time to consider my comments on their initial submission carefully. The revisions and improved imaging clarify many of my initial questions. The manuscript presents ideas about alteration of feldspars, which are volumetrically important in the crust, that are derived from observations in ore systems but equally important in granitoid crustal blocks. The manuscript reflects this importance without losing sight of the inspiration for the work, porphyry Cu systems and IOCGs, which are the world's major sources of Cu. Subjectively, I think that a high-impact paper presenting the idea that K-alteration can be driven by fluids with $Na/K > 1$, while not new here, will bring this result to a wide audience. The authors build on this to explain how complex mineralogical patterns can result from kinetic phenomena rather than a smorgasbord of crustal fluids, appealed to by many geoscientists to explain apparently multiple mineral generations. In the same way that recognition of progressive deformation debunked the idea that each 'D event' needed to be a new orogeny, so this work should make geoscientists reconsider 'how many fluids' their rocks have seen.

There are two points, which I included in the marked-up manuscript from the initial submission, that I still consider crucial to explaining the experimental data along with a concern about the significance of the work raised by recent edits. Addressing these would produce a robust and thought-provoking paper of interest to make people interested in fluid-rock interactions in crustal rocks.

1) Why does the K-feldspar precipitation begin at the surface of the grains and not at the interface where the K concentration is, presumably, higher? The recent edits (line 171-176) suggest that there is limited connection between the sanidine to albite reaction interface and the bulk of the fluid. Therefore, all the K coming out of the sanidine should be trapped here. Why then does the K-feldspar nucleate at the external surface where the fluid is less K-rich.

2) The authors describe how kinetic barriers could prevent a switch from K-feldspar precipitation to albite precipitation during reactions between magmatic fluids and porphyries (line 233-244). Can they reconcile this with their experiments that mean K-feldspar nucleation is difficult relative to albite growth even in increasingly K-rich fluids at the reaction interface. It seems like there is an equivalence here but they have not presented evidence that the relative rates of nucleation kinetics vs growth have the same relationship for both combinations of feldspar i.e. albite nucleation \ll K-feldspar growth and K-feldspar nucleation \ll albite growth.

3) Please clarify the nature of the feedback between the large-scale systems and the grain-scale process (line 250-254). If you have a change in fluid chemistry from Na to K-bearing how is this manifested at the grain scale? Similarly, how does the microscale mechanism alter the chemistry of the fluid at the large-scale? It seems that the explanation you have added potentially casts your mechanism as a second order effect, diminishing the importance of this result.

In addition, a few minor points might include the clarity of what's presented:

Line 31: Please add a reference to justify that Na is the most abundant cation in most deep fluids

Paragraph beginning line 104: Most of this paragraph is not about the reaction path modelling, just the static situation. It is very important because it illustrates how Na-rich fluids can be in equilibrium with K-feldspar. You might be advised to add a sentence along the lines of "before we do the modelling we examine the effects of Na-fluid content on predicted feldspar compositions" or something similar.

Line 133-134: Make it clear that Figure 4C is not actually a numerical model from the same software as 4A and B but essentially a summary of your experimental/analytical data.

RESPONSE TO REVIEWERS

Reviewer #2 (Remarks to the Author):

The authors have carefully addressed all the points raised in the first submission of the manuscript. I think that the manuscript should be published in its current form.

We thank the reviewer for the positive assessment of this manuscript.

Reviewer #3 (Remarks to the Author):

Review of “Kinetically driven successive sodic and potassic alteration of feldspar” by Duan et al submitted to Nature Communications.

Mark Pearce, CSIRO Mineral Resources, Australia

I am excited to read the revised version of this manuscript and thank the authors for taking the time to consider my comments on their initial submission carefully. The revisions and improved imaging clarify many of my initial questions. The manuscript presents ideas about alteration of feldspars, which are volumetrically important in the crust, that are derived from observations in ore systems but equally important in granitoid crustal blocks. The manuscript reflects this importance without losing sight of the inspiration for the work, porphyry Cu systems and IOCGs, which are the world’s major sources of Cu. Subjectively, I think that a high-impact paper presenting the idea that K-alteration can be driven by fluids with $Na/K > 1$, while not new here, will bring this result to a wide audience. The authors build on this to explain how complex mineralogical patterns can result from kinetic phenomena rather than a smorgasbord of crustal fluids, appealed to by many geoscientists to explain apparently multiple mineral generations. In the same way that recognition of progressive deformation debunked the idea that each ‘D event’ needed to be a new orogeny, so this work should make geoscientists reconsider ‘how many fluids’ their rocks have seen.

There are two points, which I included in the marked-up manuscript from the initial submission, that I still consider crucial to explaining the experimental data along with a concern about the significance of the work raised by recent edits. Addressing these would produce a robust and thought-provoking paper of interest to make people interested in fluid-rock interactions in crustal rocks.

We thank the reviewer for the positive assessment of this manuscript. We are sorry to have missed some of the earlier feedback. The marked-up manuscript in the previous round of review was not available to us on the editorial management system.

1) Why does the K-feldspar precipitation begin at the surface of the grains and not at the interface where the K concentration is, presumably, higher? The recent edits (line 171-176) suggest that there is limited connection between the sanidine to albite reaction interface and the bulk of the fluid. Therefore, all the K coming out of the sanidine should be trapped here. Why then does the K-feldspar nucleate at the external surface where the fluid is less K-rich.

In effect, in both cases (albite and K-feldspar), nucleation starts on the outside of the grain, and proceeds beyond the point of bulk saturation for each phase. This indicates that growth of the daughter mineral dominates at the reaction interface, irrespective of the K/Na mass balance at the interface. The reviewer correctly points out that the added description of the micro-porosity in the albite give the impression of limited exchange between reaction front and bulk solution. This is unintended, as reaction would stall if this connection was weak and K was actually trapped at the interface. Simple mass balance calculations show that the bulk solution will reach saturation with respect to K-feldspar after replacement of ~40% of sanidine by albite. We thank the reviewer for pointing out this important point. We have improved Fig. 3B to better show the

underlying equilibrium drivers and show the results of mass balance calculations, and we have improved the wording of this discussion:

As the reaction rim expands, chemical exchanges between the reaction front and the bulk solution occur either through (transient) reaction-induced porosity (e.g., gap between albite and sanidine during early albite precipitation; Fig. 2A)¹⁰, or in the absence of a connected porosity network, along the reaction interface^{16,17}. In either case, the conditions at the reaction interface differ from those at the outside of the grain, but relatively fast exchange of ions between the interface and bulk fluid must happen to enable the reaction to proceed³¹. (L. 170-175)

As reaction proceeds to ~40% sanidine replacement, the bulk solution reaches the peritectic composition (Fig. 3B), and the reaction is expected to stop. Instead, albitisation proceeds (Fig. 1), until eventually, K-feldspar starts replacing albite from the outside of the grains, via a new, distinct reaction interface (Fig. 2) that is completely decoupled from the albite-sanidine interface. (L.186-190)

Overall, in both cases (albite and K-feldspar), nucleation starts on the outside of the grain, and the daughter feldspar composition is consistent with the bulk fluid chemistry. However, each reaction then proceeds beyond the point of bulk saturation for each phase: albite replaces sanidine to an extent where the bulk solution becomes oversaturated with respect to K-feldspar, but undersaturated with respect to albite. This process then repeats itself, with nucleation of a K-rich feldspar replacing albite on the outside of the grain, leading to near complete replacement of the earlier formed albite. We conclude that the back-reaction responsible for K-feldspar formation is the result of kinetic processes that prevent a swap in the composition of feldspar at the reaction interface: nucleation takes place at the interface between grain and bulk fluid, but dissolution and growth dominate at the reaction interface. (L.196-205)

New ref: [31] Altree-Williams A, Pring A, Ngothai Y, Brugger J. Textural and compositional complexities resulting from coupled dissolution–reprecipitation reactions in geomaterials. Earth-Science Reviews 2015, 150: 628-651.

2) The authors describe how kinetic barriers could prevent a switch from K-feldspar precipitation to albite precipitation during reactions between magmatic fluids and porphyries (line 233-244). Can they reconcile this with their experiments that mean K-feldspar nucleation is difficult relative to albite growth even in increasingly K-rich fluids at the reaction interface. It seems like there is an equivalence here but they have not presented evidence that the relative rates of nucleation kinetics versus growth have the same relationship for both combinations of feldspar i.e. albite nucleation \ll K-feldspar growth and K-feldspar nucleation \ll albite growth.

We thank the reviewer for the positive assessment of this manuscript. We have tried to improve the clarity of this discussion – this issue indeed is linked to the point raised in (1). We now clearly indicate that nucleation of new feldspar compositions occurs only on the surface of the grains, rather than at the reaction interface; only growth (and dissolution of the parent) occur at the reaction interface. We have followed the reviewer’s recommendation and added an explicit explanation for this behaviour:

This is most likely because formation of a stable nucleus of feldspar with the new composition is statistically unlikely: atoms attaching to the growth layer edge (initial nucleation) must make two or more bonds, while only one bond is required during growth³², and the fluid contains higher Na/K ratio than the thermodynamically stable K-feldspar. (L.181-185).

New ref [32] Benning LG, Waychunas GA. Nucleation, growth, and aggregation of mineral phases: Mechanisms and kinetic controls. Kinetics of Water-Rock Interaction. Springer, 2008, 259-333.

3) Please clarify the nature of the feedback between the large-scale systems and the grain-scale process (line 250-254). If you have a change in fluid chemistry from Na to K-bearing how is this manifested at the grain scale? Similarly, how does the microscale mechanism alter the chemistry of the fluid at the large-scale? It seems that the explanation you have added potentially casts your mechanism as a second order effect, diminishing the importance of this result.

We agree that our results provide an alternative to the “externally driven” paradigm. However, we think that it is equally important to point out that the two mechanisms are not exclusive, but can work hand in hand. In effect, without the kinetic effects revealed in this paper, we would argue for albitisation to not be a pervasive process, and neither would the sequential sodic / potassic alteration be so widespread/characteristic. We have reorganised the final paragraph, starting with an introductory statement:

Our result can explain the pervasive nature of albitisation and potassic alterations, as self-evolved processes acting at grain-scale and large-scale external drivers may work together to account for the widespread distribution of successive sodic-potassic association at both micro- and macro-scales. (L.264-267)

In addition, a few minor points might include the clarity of what’s presented:

Line 31: Please add a reference to justify that Na is the most abundant cation in most deep fluids

Thanks for the suggestion. We have added a reference to suggest that in the Manuscript as following:

These fluid compositions are typical for magmatic fluids, that are originally rich in NaCl⁴⁴, as Na⁺ and Cl⁻ are the most common cation and anion in deep hydrothermal fluids⁴⁵. (L. 240-242)

New ref [45] Lecumberri-Sanchez P, Bodnar RJ. Halogen geochemistry of ore deposits: contributions towards understanding sources and processes. The role of halogens in terrestrial and extraterrestrial geochemical processes. Springer, 2018, 261-305.

Paragraph beginning line 104: Most of this paragraph is not about the reaction path modelling, just the static situation. It is very important because it illustrates how Na-rich fluids can be in equilibrium with K-feldspar. You might be advised to add a sentence along the lines of "before we do the modelling we examine the effects of Na-fluid content on predicted feldspar compositions" or something similar.

Thanks for the suggestion. We have added a sentence in the beginning of that paragraph in the Manuscript as following:

First, we examine the fractionation of Na and K between feldspar and hydrothermal solution under static thermodynamic equilibrium conditions. All thermodynamic calculations were performed with GEM-Selektor software package, as it has been demonstrated that this package can effectively model such complex thermodynamic conditions^{26,27} (details in the Supplementary Data). (L.104-107)

Line 133-134: Make it clear that Figure 4C is not actually a numerical model from the same software as 4A and B but essentially a summary of your experimental/analytical data.

Thanks for the suggestion. We have emphasized the Figure 4C is a summary of this study in the Manuscript as following:

Our experiments, however, display different final product compositions and textures (as summarized in Fig. 4C) than those predicted by equilibrium thermodynamic modelling (Figs. 4A, B). (L.136-138)

REVIEWERS' COMMENTS

Reviewer #3 (Remarks to the Author):

Review of "Kinetically driven successive sodic and potassic alteration of feldspar" by Duan et al

Mark Pearce, CSIRO Mineral Resources, Australia

The authors have addressed the previous comments made, most importantly regarding why the K-feldspar nucleates at the grain margins and not at the interface where conceptually one might expect the highest K concentrations during sanidine dissolution. The mass balance showing that, even though the bulk solution reaches peritectic composition at 40% sanidine dissolution, the reaction continues adds to the argument that the microstructures are kinetically controlled. This is, for me, one of the the aspects of the paper that should influence thinking about alteration in fluid-rock reaction systems, namely that continued precipitation of now metastable minerals is sometimes much easier than nucleating new phases.

The significance of the sequential, kinetically drive replacement of albite by K-feldspar is certainly widespread in natural systems. The authors approach to presenting their hypothesis as a complementary, potentially coincident mechanism to changing external fluid compositions or conditions allows a full range of alternatives to be considered by workers in the field in future.

As with previous versions of the manuscript, the figures are clear, well drafted and pertinent to the narrative being developed. Descriptions of methods and analysis techniques will allow the results to be reproduced.

I look forward to seeing this paper published in the near future.